# Associations of the Single Bovine Embryo Growth Media Metabolome with Successful Pregnancy

**DOI:** 10.3390/metabo14020089

**Published:** 2024-01-26

**Authors:** Elina Tsopp, Kalle Kilk, Egon Taalberg, Pille Pärn, Anni Viljaste-Seera, Ants Kavak, Ülle Jaakma

**Affiliations:** 1Chair of Animal Breeding and Biotechnology, Institute of Veterinary Medicine and Animal Sciences, Estonian University of Life Sciences, 51006 Tartu, Estonia; pille.parn@emu.ee (P.P.); anni.viljaste-seera@emu.ee (A.V.-S.); ylle.jaakma@emu.ee (Ü.J.); 2Department of Biochemistry, Institute of Biomedicine and Translational Medicine, University of Tartu, 50411 Tartu, Estonia; kalle.kilk@ut.ee (K.K.); egon.taalberg@ut.ee (E.T.); 3Chair of Clinical Veterinary Medicine, Institute of Veterinary Medicine and Animal Sciences, Estonian University of Life Sciences, 51006 Tartu, Estonia; ants.kavak@emu.ee

**Keywords:** bovine, embryo, metabolomics, in vitro, pregnancy, biomarkers

## Abstract

This study investigated whether metabolomic fingerprints of bovine embryo growth media improve the prediction of successful embryo implantation. In this prospective cohort study, the metabolome from in vitro-produced day 7 blastocysts with successful implantation (*n* = 11), blastocysts with failed implantation (*n* = 10), and plain culture media without embryos (*n* = 5) were included. Samples were analyzed using an AbsoluteIDQ^®^ p180 Targeted Metabolomics Kit with LC-MS/MS, and a total of 189 metabolites were analyzed from each sample. Blastocysts that resulted in successful embryo implantation had significantly higher levels of methionine sulfoxide (*p* < 0.001), DOPA (*p* < 0.05), spermidine (*p* < 0.001), acetylcarnitine-to-free-carnitine ratio (*p* < 0.05), C2 + C3-to-free-carnitine ratio (*p* < 0.05), and lower levels of threonine (nep < 0.001) and phosphatidylcholine PC ae C30:0 (*p* < 0.001) compared to control media. However, when compared to embryos that failed to implant, only DOPA, spermidine, C2/C0, (C2 + C3)/C0, and PC ae C30:0 levels differentiated significantly. In summary, our study identifies a panel of differential metabolites in the culture media of bovine blastocysts that could act as potential biomarkers for the selection of viable blastocysts before embryo transfer.

## 1. Introduction

Fertility in cattle has decreased dramatically in recent decades and is currently one of the major challenges in the dairy industry [1]. Assisted reproductive technologies (ART) play a crucial role in addressing low fertility and have been widely used in cattle worldwide to enhance the reproduction of animals with high genetic value [2,3]. Moreover, bovine in vitro embryo production (IVEP) enables the use of sub-fertile donors and repeat breeders. IVEP has evolved into a significant international business, with over a million transfers of in vitro-produced bovine embryos recorded worldwide [4].

Successful conception, embryo survival, and the birth of a healthy calf are major factors influencing the production and economic efficiency of a beef or dairy farm. Several conditions are known to lead to implantation failure after embryo transfer (ET), including the health status of the recipient animal, reduced endometrial receptivity, genetic abnormalities of an embryo, errors in the embryo transfer technique, or a combination of these influences [5]. One of the first essential steps in achieving a successful pregnancy is the selection and transfer of highly competent in vitro-produced embryos, which is critical for the application of IVF technology in a large animal breeding toolbox.

There are currently three main trending research areas in the embryo industry: time-lapse microscopy empowered by AI-based computational modelling [6]; the analysis of extracellular vesicles (EVs) secreted by blastocysts [7]; and, lastly, the correlation of metabolomic/proteomic markers in spent culture medium to embryo viability [8,9,10,11].

In practice, however, the most prevalent method to assess bovine embryos’ developmental potential in vitro is to determine the cleavage rate paired with morphological assessment, which, sadly, is a highly subjective method based on the experience of the embryologist [12]. Even though the first calf after the transfer of an IVEP embryo was born 40 years ago, the selection of viable IVEP embryos in the livestock industry is still insufficient, and the industry is missing a reliable, non-invasive, and cost-effective technique to assess embryos before ET [13]. Low IVEP embryo transfer success rates inevitably bring a higher overall cost per calf born.

Despite a considerable amount of promising research results, a practical and accurate non-invasive assessment system has not yet been developed for the cattle industry [14]. The need for a fast, precise, and practical method of determining embryo viability and predicting its ability to implant has driven years of research directed at finding the most suitable method for clinical application.

The outstanding advances in non-invasive metabolomic analysis in recent years, enabling the assessment of single embryo physiology, are now swiftly becoming a part of everyday work in human IVF labs and will hopefully soon be applied in bovine IVF labs as well. Changes in embryo metabolism during early development mirror cellular metabolic activities and the overall viability of the embryo [15]. As the metabolome reflects the end-products of embryos’ physiological activities, our primary objective in this study was to evaluate the metabolome through metabolomics and explore its relationship with successful pregnancy outcomes. This approach enhances our comprehension of the health status in pre-implantation bovine embryos.

Given this robust background, the present study aimed to test the effectiveness of targeted metabolomic analysis using the AbsoluteIDQ^®^ p180 Targeted Metabolomics Kit coupled with liquid chromatography-tandem mass spectrometry (LC-MS/MS). This approach allows for the comprehensive capture of a broad range of molecular content in individually cultured bovine embryo culture media. We hypothesized that metabolomic profiles of embryos with successful implantation differ from those of embryos with failed implantation. To the best of our knowledge, this study is the first of its kind to utilize the AbsoluteIDQ^®^ p180 Targeted Metabolomics Kit to determine metabolomic profiles of individually cultured bovine IVEP embryos, aiming to identify metabolomic signatures associated with successful implantation and those associated with failed implantation. A non-invasive approach for predicting embryo viability could offer multiple benefits, reducing the risk of impairing the embryo and its environment, thereby maintaining development potential. Additionally, it has the potential to lower the overall cost of IVEP and embryo transfer by contributing to higher pregnancy rates and reducing the number of open recipients [16].

## 2. Materials and Methods

### 2.1. Media

Serum-free media for every step of in vitro embryo production, including in vitro maturation, fertilization, and the individual cultivation of embryos, were obtained from IVF Limited T/A IVF Bioscience (Bickland Industrial Park, Falmouth, Cornwall, UK), unless stated otherwise. Care was taken to select media for all experiments from the same production batch.

### 2.2. Experimental Design

We embraced embryos produced from OPU-derived oocytes in our study to enable the transfer of such embryos and to compare the metabolic fingerprints of blastocysts resulting in pregnancy with those that did not. The culture media samples were collected prior to embryo transfer on day 7 of culture and were derived from IETS Grade 1 blastocysts with confirmed pregnancy (ET-P BL; *n* = 11), IETS Grade 1 blastocysts with failed implantation (ET-NP BL; *n* = 10), and plain growth media (CM) for control (*n* = 5), resulting in a total of 26 culture media samples.

Transvaginal ultrasound-guided follicular puncture (Ovum Pick-Up; OPU) and in vitro embryo production were employed to generate blastocysts for embryo transfers. Holstein donor cows, pregnant between 45 and 100 days of gestation, and non-pregnant Holstein donor cows, 40 to 90 days after parturition, were utilized in the OPU procedure. No superovulation of donor cows was used before OPU. Epidural anesthesia was administered using Xylazine (0.05 mg/kg body weight). Ovaries were scanned with a 7.5 MHz micro-convex transducer, and follicles ≥ 3 mm were punctured using an 18 G needle coupled to the aspiration device and vacuum system. Follicular fluid, mixed with BO-OPU media, was transported to the laboratory within one hour.

Cumulus–oocyte complexes (COCs) with three or more layers of unexpanded cumulus cells and morphologically bright, evenly granulated cytoplasm were selected for in vitro maturation. COCs were washed thrice in HEPES buffered medium (BO-Wash) and once with maturation medium (BO-IVM). COCs from each donor cow were matured separately in four-well plates (Nunclon, Roskilde, Denmark), each containing 500 µL BO-IVM medium. Oocytes were incubated at 38.5 °C with 5% CO_2_ in humidified air for 24 h.

### 2.3. In Vitro Fertilization

Frozen-thawed semen (commercially produced at Animal Breeders’ Association of Estonia, Keava) from a single bull was prepared by thawing straws in water at 35 °C for 30 s. In vitro fertilization (IVF) was conducted using semen from the bull EHF ZIARD 27481 (ID EE13993023). Semen from one straw was gently expelled into a 15 mL centrifuge tube containing 3 mL of BO-SemenPrep medium and centrifuged for 5 min at 328× *g* at 23 °C. The supernatant was removed, 3 mL of BO-SemenPrep was added, and the same centrifugation process was repeated. Matured oocytes were washed once in BO-IVF medium and placed in four-well plates with 500 µL BO-IVF medium per well. The cumulus–oocyte complexes (COCs) and sperm were co-incubated at a final concentration of approximately 1 × 10^6^ motile sperm/mL at 38.5 °C with 5% CO_2_ in air with maximum humidity for 18 h.

### 2.4. In Vitro Cultivation and Embryo Transfer

Presumptive zygotes were denuded 18 h after IVF by vortexing for 120 s in BO-Wash medium, rinsed thrice in BO-Wash medium, and once in BO-IVC medium. Zygotes were cultured in 60 µL BO-IVC culture medium droplets (one zygote per droplet) overlaid with mineral oil in 100 mm Petri dishes (Sigma-Aldrich, St. Louis, MO, USA) at 38.5 °C under an atmosphere of 5% CO_2_, 5% O_2_, and 90% N_2_ for up to 7 days. Controls were created by placing 60 µL of plain BO-IVC culture media droplets that were never in contact with an embryo under oil, and they were kept under the same conditions as zygotes (38.5 °C under an atmosphere of 5% CO_2_, 5% O_2_, 90% N_2_). Fresh day 7 IETS Grade 1 blastocysts were transferred to the synchronized EHF recipient heifers using BO-Transfer media.

### 2.5. Collection of Media for LC-MS/MS and Categorization of Samples

Culture media samples (20 µL) were collected from droplets of individually cultured OPU-derived day 7 blastocysts. Morphological assessments were conducted on each embryo at the time of media collection, adapted from previously published work [17,18]. Any embryos that were arrested in development by day 7 were excluded from the experiment. Subsequently, samples were categorized according to their pregnancy outcomes: “ET-P” for blastocysts resulting in pregnancy and “ET-NP” for blastocysts that failed to result in pregnancy at day 30 following embryo transfer. Plain media samples were also collected on day 7 and processed as controls (hereafter referred to as “Control”), following the approach adopted from previous research [19]. All collected media samples were labeled and stored immediately after collection at −20 °C.

### 2.6. Preparation of Culture Media Samples for LC-MS/MS and Spectrometry

Each 20 µL culture media sample was prepared following The AbsoluteIDQ^®^ p180 Targeted Metabolomics Kit sample preparation protocol for Agilent Infinity high-performance liquid chromatography (Agilent, Santa Clara, CA, USA) coupled to the 4500 QTRAP^®^ ion trap mass spectrometer (Sciex, Framingham, MA, USA).

### 2.7. Statistical Analysis

Based on the raw chromatography-mass spectrometry data, all missing values were treated as such due to being below the detection limit and were subsequently substituted with zero. Univariate comparisons of each metabolite or derived indexes (metabolite sums or ratios) between different media samples were conducted. Metabolites with more than 30% missing values were compared using the chi-squared test to determine whether the metabolite was significantly more frequently detectable in any of the study groups. If less than 30% of the values were missing, the Shapiro–Wilk test was used to assess whether the data resembled a normal distribution. Depending on this test, either ANOVA with a Tukey’s honestly significant post-hoc or Kruskal–Wallis with a Dunn post-hoc test were applied. The results were considered statistically significant at *p* < 0.001, and *p*-values between 0.05 and 0.001 were considered borderline significant. All statistical analyses were performed and all figures were constructed using R version 4.2.0 (R Foundation for Statistical Computing, Vienna, Austria).

## 3. Results

A total of 26 spent media samples were analyzed using the AbsoluteIDQ^®^ p180 Targeted Metabolomics Kit with LC-MS/MS, covering a total of 189 metabolites: 40 acylcarnitines, 42 amino acids/biogenic amines, 91 phospholipids, 15 sphingolipids, and the sum of hexoses (Appendix A). The culture media samples were derived from IETS Grade 1 day 7 blastocysts with confirmed pregnancy (*n* = 11), IETS Grade 1 day 7 blastocysts with failed implantation (*n* = 10), and from day 7 plain growth media for control (*n* = 5).

### 3.1. Metabolites Differing in Culture Media of Blastocysts and Controls

In total, 8 metabolites showed significant differences (*p* < 0.001) and 27 exhibited borderline significance (*p* between 0.05 and 0.001) between the blastocyst culture media and empty controls (Table 1). Spermidine was not detectable in the empty media (control), but was found in most of the media from pregnancy-yielding blastocysts and in a few media samples from blastocysts without pregnancy.

Among amino acids, only threonine (Thr) was significantly consumed by blastocysts. Glutamine (Gln) was secreted into the culture media (*p* = 0.007), and dihydroxyphenylalanine (DOPA), a metabolite of phenylalanine (Phe) and tyrosine (Tyr), was detectable in 64% of ET-P samples and 20% of ET-NP culture media samples, while being completely absent in empty media. Additionally, oxidized methionine (Met-SO) levels rose in media containing blastocysts.

Among acylcarnitines, methylglutaryl-(C5M-DC) and decanoylcarnitines (C10) were increased in media containing blastocysts. Interestingly, pimeloylcarnitine (C7-DC) levels were reduced in media containing blastocysts (Table 1). Blastocysts secreted carnitine esters with C4:1, C4, C5, C18:2, C5:1-dicarboxylic acid, and hydroxylated C14:1-OH, C16:1-OH, C18:1-OH fatty acid residues into the culture media with borderline significances.

Thirteen phosphatidylcholines (PCs) out of the measured seventy-six and three lysophosphatidylcholines (lysoPCs) out of the fourteen measured ones had at least borderline significance in a groupwise comparison. Both increases and decreases in their concentration in culture media were observed due to blastocyst presence (Figure 1). PC ae C30:0 and PC aa C28:1 in culture media were significantly depleted by blastocysts, whereas the concentration of PC ae C30:3 was distinct in all three experimental groups.

### 3.2. Derived Indicators Differing between Culture Media of Blastocysts and Controls

Metabolites of the same class share metabolic pathways. Summary concentrations of these, or ratios of a pathway’s start and end product or enzyme substrate and product, may be more informative than individual metabolites. Forty-four sums or ratios of metabolites were calculated (Appendix A). The ratio of oxidized methionine to methionine was the only calculated indicator of metabolic activity that significantly differentiated control media from media containing blastocysts (Table 2). The summary concentration of all PCs, mainly due to PC aa type lipids and saturated fatty acid residues, had borderline significance. The relative amount of dicarboxylic acid esters to all carnitine esters also had borderline significance between the groups.

### 3.3. Metabolites Differing between Media of Pregnancy-Yielding and Unyielding Blastocysts

Post-hoc analysis revealed significant differences in PC ae C30:0 concentrations among all three groups (Table 1). Among the metabolites with borderline significance, PC aa C42:1, PC ae C34:3, and linoleic acid carnitine ester differentiated in the ET-P BL and ET-NP BL groups. All PC-s had a lower concentration in the ET-P BL group, and the carnitine ester had a higher concentration in the culture media of the ET-P BL group. The total concentration of PC aa was significantly lower in the culture media of pregnancy-yielding blastocysts than in the ET-NP BL group and controls (Table 2). The acetylcarnitine-to-free-carnitine ratio was significantly higher in the ET-P BL group than in either of the other groups.

## 4. Discussion

Maximizing pregnancy outcomes relies on the non-invasive assessment of blastocysts, and the metabolomic characterization of pre-implantation growth media has shown some success. Although various metabolites have been proposed as embryo viability markers, the results are scarce and reproducibility seems to be a challenge. Certainly, the development and putative markers may be influenced by the experimental setup and many other factors [20].

### 4.1. Glucose Metabolism

The dominant hexose in growth media is glucose, and, as expected, both ET-P BL and ET-NP BL blastocysts consumed a fraction of it in our experiment. Glucose serves as a substrate for ATP and NADPH production once the biosynthesis gears up after the initial embryo cleavages [21]. In some studies, the level of glucose consumption has been linked to the quality and success of pregnancy [22]. However, the glucose usage intensity of embryos was not correlated with pregnancy in our results.

### 4.2. Amino Acid Metabolism

Amino acids are utilized in the de novo synthesis of nitrogen-containing compounds (e.g., nucleobases and heme) and lipids (serving as precursors for sphingolipids and choline), and, of course, proteins. Additionally, they can serve as a source of ATP. Essential amino acids cannot be synthesized by mammals; thus, their levels are expected to decrease in growth media. Glutamate (Glu), Gln, and Ala are central in transamination, thereby linking the metabolisms of nitrogen-containing and nitrogen-free metabolism. Depending on blastocyst activity, their levels in growth media may also increase, even if nitrogen waste is in other forms [11].

It has been previously shown that the consumption of essential amino acids can vary among embryos. Two decades ago, it was discovered that embryos require different amino acids at various developmental stages [23,24]. Gln, arginine (Arg), Met, Ala, asparagine (Asn), serine (Ser), and leucine (Leu) were highlighted then. However, there were disagreements between the early reports, possibly due to suboptimal culture conditions such as oxygen levels in incubators and the absence of serum in culture media, which may have led to erroneous conclusions [25].

In a recent bovine pregnancy prediction study [26], it was found that Glu, proline (Pro), Met, Arg, lysine (Lys), and Thr concentrations in embryo culture media have the best pregnancy-predicting power, but only when a single metabolite was used in the assessment. Lechniak et al. [10] compared different culture setups and found distinct differences in the metabolic pathways of Phe, Tyr, Met, aspartate (Asp), Arg, Pro, and histamine (His) with regard to the culture system used. In the same study, cysteine (Cys), Ala, Glu, glycine (Gly), Ser, Phe, Tyr, Lys, and Asp-dependent pathways appeared to be associated with the zygotic cleavage pattern, serving as potential viability indicators. While these lists may seem extensive, many amino acids are directly related to each other; for example, Phe and Tyr probably mark a single process. Ser and Gly can be converted to each other, potentially affecting Met levels. Ser and Gly also relate to Ala and Thr in one- or two-step conversions [27].

In our study, most of the amino acids did not differ significantly between media containing blastocysts and empty control media, indicating that the consumption and secretion of amino acids by embryos were relatively modest compared to their levels in growth media. This aligns with the theory that viable bovine embryos exhibit a “quiet” metabolism rather than an “active” one, although the limits of this quietness are yet to be defined, and there is still much to be learned [28]. Similarities between the successful and failed implantation groups could also be attributed to the selection of embryos with similar morphological qualities for embryo transfer. Interestingly, Thr, an essential amino acid, was a striking exception. Thr levels were significantly lower in media from successfully implanted embryos relative to empty control media; however, we did not observe significant differences between the successful and failed implantation groups. This finding is consistent with other recent works [8].

Whether Thr is indeed more special than other amino acids remains to be confirmed, but one theory suggests that threonine dehydrogenase (TDH) might be key for bovine blastocyst development [29]. This enzyme uses Thr to produce a one-carbon fragment and reduced NADH. Both products can be further used in anabolic processes. Met, a usual donor of one-carbon fragments for methylation reactions, can, thus, be reserved for other uses, such as sulfur-based components of the antioxidant system.

Thr plays an important role in the growth and proliferation of embryonic stem cells (ES) in pre-implantation embryos, where it is converted to Gly and acetyl-CoA by TDH. Gly and acetyl-CoA aid nucleotide biosynthesis, epigenetic modifications, and mitochondrial free energy conversions in ES cells [30].

In relation to Met, an essential amino acid whose metabolism has been prominently featured in recent blastocyst quality assessments and embryo implantation studies [31], it is noteworthy that numerous studies have confirmed an increase in the concentration of Met in viable embryos [32], as well as in the uterine lumen of pregnant animals [33,34]. Met is oxidized to Met-SO, and its levels, which were significantly different in our study, are considered to protect cells from oxidative damage, although a more complex bioregulatory role cannot be excluded [35]. Specifically, we observed that Met-SO levels were significantly higher in media from successfully implanted embryos and in the failed implantation group, relative to the empty control media.

Aromatic amino acids did not exhibit any significant distinctions, except for DOPA, which was unexpectedly measurable in 64% of samples from embryos with successful pregnancies and in 20% of samples from embryos with implantation failure, while being completely absent in empty culture media. This suggests that DOPA is synthesized and secreted into the culture media by embryos. To date, no study has demonstrated the metabolism of DOPA or its role in early-developing bovine embryos. However, it is well-known that catecholamine synthesis in the fetus begins at mid-gestation with the conversion of Tyr into 3,4-dihydroxyphenylalanine, mediated by the enzyme tyrosine hydroxylase (TH). DOPA can subsequently be converted to dopamine by the aromatic L-amino acid decarboxylase [36]. While previous studies have provided little evidence for any role of DOPA in early embryo development, further research is needed to investigate this peculiar finding. Catecholamine precursors such as Phe and Tyr or related compounds are commonly reported in metabolomic studies [9,10]. The role of catecholamines in blastocyst development has also been suggested by Wang et al. [36]. There are hints that catecholamines might regulate lipid usage [37] and the synthesis of cell differentiation-regulating polyamines [38], but this knowledge needs to be explored further before definitive conclusions can be drawn.

### 4.3. Polyamine Metabolism

The most outstanding difference in our study between culture media of pregnancy-yielding and non-yielding blastocysts was spermidine, a polyamine. Spermidine is a polycationic substance synthesized from Arg and Met and is known to regulate embryonic gene expression and protein synthesis, playing a crucial role in the growth and proliferation of embryos [39]. The essential role of spermidine in developing blastocysts has been reported previously [40,41]. In our experiment, spermidine was not detectable in empty media (control), but it was found in 90.91% of media from successfully implanted blastocysts and in 20% of media from embryos with failed implantation. Therefore, spermidine could be a potential biomarker for selecting viable bovine embryos for embryo transfer, as it appears to be closely related to successful implantation and pregnancy. Several gaps in our knowledge around spermidine metabolism in early bovine blastocysts could benefit from further research, including larger trial groups of transferable embryos and a higher number of embryo transfers and recipients.

### 4.4. Lipid Metabolism

Most of the changes observed in any comparison among the three growth media groups were related to lipids. The multifaceted roles of lipids in early embryo development were recently reviewed by Melo-Sterza and Poehland [20]. The beta-oxidation and carnitine esterification of fatty acids emerge as particularly crucial aspects of embryo lipid metabolism [42]. In an experimental study on bovine pregnancy prediction, Gomez et al. [9] identified free fatty acids C10, C16, C18, and C18 glycerol ester as potential viability markers in later stages of embryo development. Although Lechniak et al. [10] did not report individual metabolites for embryo selection, they found that the beta-oxidation of fatty acids seems highly dependent on embryo culture conditions, providing insight into the embryo’s response to environmental factors.

To provide a broader perspective beyond individual lipid species, we observed that media containing blastocysts in our study were richer in oxidized (DC- dicarboxylic and OH- hydroxylated) fatty acid residues, as well as short-chain fatty acids (SCFA). This indicates the active oxidation (beta or, in the case of DC variants, omega oxidation) of fatty acids during embryo development. When comparing the ET-P and ET-NP groups, differences in the concentration of long-chain acylcarnitines in culture media were noted, along with acetylcarnitine and the C2 + C3-to-free-carnitine ratio, which was significantly higher in the culture media of blastocysts with successful implantation. The acetyl group and long-chain fatty acids mark the beginning and end of fatty acid synthesis in one direction and beta-oxidation in the other. Higher levels of their carnitine esters suggest more efficient ester synthesis. Carnitine supplementation during in vitro culture is known to improve embryo potential for implantation and successful pregnancy [43]. The stimulatory effect of carnitine on lipid metabolism has been confirmed in cattle, mice, and pig oocytes [44]. L-carnitine, a powerful antioxidant, reduces the accumulation of reactive oxygen species (ROS) and decreases apoptosis in animal cells [45]. Our results suggest that it is not just the level of carnitine itself, but the efficiency of its use—such as the acetylcarnitine-to-free-carnitine ratio and C2 + C3-to-free carnitine ratio—that might be a decisive factor for blastocyst quality and subsequent implantation.

### 4.5. Phosphatidylcholine Metabolism

PCs play a crucial role in membrane synthesis during cell division and can also function as substrates for beta-oxidation, as well as storage for polyunsaturated fatty acids, which serve as precursors for signaling molecules [46]. PCs, being dominant membrane phospholipids, play a vital role in maintaining the structural integrity of the lipid bilayer during embryonic development [47].

In our study, there was a significant difference in PC ae C30:0 levels in culture media between blastocysts with successful and failed implantation. PC ae C30:0 is a saturated plasminogen. Additionally, the concentration of PC ae C30:3 was significantly lower in culture media of blastocysts with successful implantation compared to the ET-NP and empty culture media groups. Notably, the sum of all saturated fatty acid-containing PCs exhibited a similar trend, with lower concentrations in the ET-P group and higher concentrations in the ET-NP group.

While the biological function of PC ae C30:0 in early embryo development remains undescribed, our study is the first to investigate PC ae C30:0 levels in culture media and their association with viable embryos. Although studies in human plasma have explored PC ae C30:0 concentration and its correlation with conditions such as ovarian endometriosis and prostate cancer [48], further replication of our findings is essential before PC ae C30:0 can be considered a biomarker for selecting blastocysts with the highest implantation potential.

### 4.6. Future Research Directions

In the current study, oocytes for embryo production were obtained from both pregnant and non-pregnant donor cows, a factor known to influence oocyte quality and blastocyst rates, as indicated in previous studies [49]. Specifically, cleavage and blastocyst rates are higher when using oocytes from pregnant donor cows. However, in the current study, we did not investigate the metabolomic differences of embryos obtained from either pregnant or non-pregnant donors. This intriguing aspect could be explored in future studies, as there may be metabolic differences in the culture media of embryos depending on the oocyte donor’s reproductive status.

The current study focused on metabolomic analysis exclusively in day 7 blastocysts of excellent quality (IETS Grade 1) for the initial validation of target metabolites, given that Grade 2 embryos yield lower pregnancy rates than Grade 1 embryos [50]. In field conditions, the transfer of Grade 2 embryos is feasible, albeit with diminished pregnancy outcomes [51]. In the dairy cattle industry, avoiding the transfer of Grade 2 embryos to lactating cow recipients is common, as these animals must achieve pregnancy shortly after calving to maximize economic returns, whereas transferring Grade 2 embryos to heifers is generally accepted [52]. Conducting a preventive analysis of target metabolic pathways in the culture media of Grade 2 embryos could optimize overall ET efforts and should be explored in future studies.

## 5. Conclusions

In conclusion, our study affirms the significance of lipid beta-oxidation, Met, aromatic amino acids, and polyamine metabolisms as pivotal processes in blastocyst preimplantation development. When considering our findings alongside the existing literature, these pathways emerge as the most promising candidates for providing biomarkers to predict blastocyst viability and assess the embryo’s potential for successful implantation. While individual metabolites may vary across studies, the core processes likely remain consistent, with the specific metabolite identified as a biomarker influenced by experimental conditions and culture media. Undoubtedly, our unexpected revelations regarding DOPA, spermidine, Met-SO, C2/C0, (C2 + C3)/C0, and PC ae C30:0 metabolism and their association with successfully implanted bovine embryos warrant careful consideration and further investigation.

## Figures and Tables

**Figure 1 metabolites-14-00089-f001:**
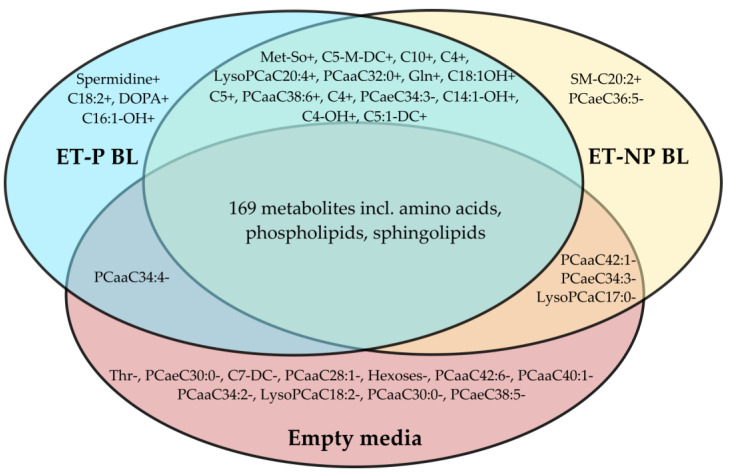
A Venn diagram summarizing the metabolites shared or differentiating among growth media of pregnancy-yielding blastocysts (“ET-P BL”), blastocysts that did not result in a successful pregnancy (“ET-NP BL”), and empty culture media. “+” highlights an increase in metabolite concentration in culture media. “-” highlights a decrease in metabolite concentration in culture media.

**Table 1 metabolites-14-00089-t001:** Metabolites exhibiting a significant difference in concentrations (indicated by a bolded *p*-value) or those displaying borderline significance. In instances where metabolites were detected in only a limited number of samples, the chi-squared test was employed and the count of samples with detectable concentrations was provided. For metabolites demonstrating an approximately normal distribution, the analysis involves ANOVA with a Tukey’s HSD post-hoc test. Conversely, for metabolites featuring a non-normal distribution of values, the Kruskal–Wallis test, coupled with a Dunn post-hoc test, was utilized.

Metabolites	Concentrations in Culture Media, μM			Significance	
Metabolite	Pregnancy	No Pregnancy	Control Media	*p*-Value	Test
Thr ↑	113.5 (±12.7) ^b^	121.7 (±12.9) ^b^	205.7 (±43.5) ^a^	**1.20 × 10^−7^**	ANOVA
PC ae C30:0 ↓	0.029 (±0.003) ^b^	0.033 (±0.003) ^c^	0.041 (±0.005) ^a^	**6.70 × 10^−6^**	ANOVA
C7-DC ↓	0.013 (±0.002) ^b^	0.015 (±0.002) ^b^	0.02 (±0.003) ^a^	**3.20 × 10^−5^**	ANOVA
Met-SO ↑	1.1 (±0.26) ^b^	0.98 (±0.28) ^b^	0.33 (±0.28) ^a^	**7.10 × 10^−5^**	ANOVA
PC aa C28:1 ↓	0.008 (±0.003) ^b^	0.009 (±0.003) ^b^	0.015 (±0.001) ^a^	**0.00019**	ANOVA
Spermidine ↑	10 (90.91%)	2 (20%)	0 (0%)	**0.00035**	Chi-square
C5M-DC ↑	0.042 (±0.004) ^b^	0.04 (±0.003) ^b^	0.032 (±0.005) ^a^	**0.00038**	ANOVA
C10 ↑	0.114 (±0.013) ^b^	0.106 (±0.006) ^b^	0.091 (±0.008) ^a^	**0.00095**	ANOVA
DOPA ↑	7 (64%)	2 (20%)	0 (0%)	0.021	Chi-square

^a b c^ Culture media groups that do not exhibit statistical differences from each other were assigned the same letter, while groups with differing letters indicate significant differences between them. ↑ highlights an increase in metabolite concentration in culture media containing embryos compared to the control. ↓ highlights a decrease in metabolite concentration in culture media containing embryos compared to the control. Bolded *p*-values are significant also after multiple comparison correction.

**Table 2 metabolites-14-00089-t002:** Metabolite sums and ratios exhibiting significant differences (indicated by bolded *p*-values) or borderline significance in concentrations were examined. In cases where metabolites were detected in only a few samples, the chi-squared test was employed, providing the count of samples with detectable concentrations. For metabolites displaying a roughly normal distribution, ANOVA with a Tukey’s HSD post-hoc test was applied. In instances of metabolites featuring a non-normal distribution of values, the Kruskal–Wallis test with a Dunn post-hoc test was utilized.

Metabolites	Concentrations in Culture Media			Significance	
Metabolite	Pregnancy	No Pregnancy	Control Media	*p*-Value	Test
Met-SO/Met ↑	0.022 (±0.005) ^b^	0.02 (±0.007) ^b^	0.006 (±0.005) ^a^	**1.00 × 10^−4^**	ANOVA
SFA/PC ↓	0.76 (±0.017) ^b^	0.77 (±0.024) ^b^	0.80 (±0.015) ^a^	0.0011	ANOVA
Total PC/SM ↓	1.77 (±0.057) ^b^	1.77 (±0.043) ^b^	1.853 (±0.035) ^a^	0.0083	ANOVA
Total PC aa ↓	0.88 (0.86–0.9 ^a^) ^b^	0.9 (0.89–0.92) ^b^	0.97 (0.94–0.98) ^a^	0.00904	Kruskal
Total PC ↓	1.69 (±0.063) ^b^	1.70 (±0.043) ^b^	1.78 (±0.03) ^a^	0.013	ANOVA
C2/C0 ↑↓	0.17 (0.17–0.19) ^a^	0.16 (0.15–0.17) ^b^	0.17 (0.16–0.17) ^b^	0.0186	Kruskal
(C2 + C3)/C0 ↑↓	0.21 (0.20–0.22) ^b^	0.2 (0.19–0.21) ^a^	0.2 (0.199–0.211) ^a^	0.0285	Kruskal
Total AC-DC/Total AC ↓	0.19 (±0.008) ^b^	0.19 (±0.005) ^b^	0.198 (±0.006) ^a^	0.043	ANOVA

^a b^ Culture media groups that do not exhibit statistical differences from each other were assigned the same letter, while groups with differing letters indicate significant differences between them. ↑ highlights an increase in metabolite concentration in culture media containing embryos compared to the control. ↓ highlights a decrease in metabolite concentration in culture media containing embryos compared to the control. Bolded *p*-values are significant also after multiple comparison correction.

## Data Availability

The data presented in this study are available on reasonable request from the corresponding author. The data are not publicly available due to privacy and/or ethical concerns.

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
