# Peer review of "Associations of the Single Bovine Embryo Growth Media Metabolome with Successful Pregnancy"

_metabolites, 2024, doi:10.3390/metabo14020089_

Round 1

Reviewer 1 Report

Comments and Suggestions for Authors

The paper is well written and presents novel and interesting data. Only minor addition is necessary before deciding on publication:

2.2 Please describe in detail the experimental groups and add the exact number of studied samples per group

Lines 169 - 171: same as above, add ‘n’ for each group

Author Response

Dear Reviewer,
I appreciate your valuable suggestions and the prompt review of the manuscript. Thank you.

2.2 The experimental groups are now described in more detail and the exact number of studied samples per group is added. 

Lines 169 - 171: same as above, added ‘n’ for each group.
Please see the attachment. Changes are highlighted in yellow.

Reviewer 2 Report

Comments and Suggestions for Authors

The paper entitled "Associations of the Single Bovine Embryo Growth Media Metabolome with Successful Pregnancy" describes results from metabolomic investigation of in vitro produced Day 7 blastocysts. This work compares metabolomic from embryos which displayed successful implantation, implantation failure and  plain culture media, respectively. Results highlighted differences in metabolic pathways which can improve future selection of embryos for transfer, thus ameliorating the success rate of ET in the field. 

The manuscript is well written, the M&M well organized, while discussion of results obtaind does scientifically sound. 

Since I do believe that results from this work could be applied to the field practice in the future, I have few minor suggestions, mainly for the discussion/results sections, as reported below:

- The Authors stated that they analyzed metabolomics only from Day 7 blastocysts of quality 1 class. I agree with this choice for the initial validation of the target metabolites - before field application. However, I suggest the Authors to discuss this approach: in field conditions, class-2 enbryos could also be transferred and preventive analysis of target metabolic pathways could optimize the overall ET-efforts.

- Figure 1: Authors can evaluate whether adding "+" or "-" symbols in order to highlight which metabolite increases or dicreases, respctively. Not mandatory, but maybe this could be useful to the reader.

Author Response

Dear Reviewer,
I appreciate your valuable suggestions and the prompt review of the manuscript. Thank you.

Discussion has been updated accordingly: The current study focused on metabolomic analysis exclusively in Day 7 blastocysts of excellent quality (IETS Grade 1) for the initial validation of target metabolites, given that Grade 2 embryos yield lower pregnancy rates than Grade 1 embryos [21]. In field conditions, the transfer of Grade 2 embryos is feasible, albeit with diminished pregnancy outcomes [22]. In the dairy cattle industry, avoiding the transfer of Grade 2 embryos to lactating cow recipients is common, as these animals must achieve pregnancy shortly after calving to maximize economic returns, whereas transferring Grade 2 embryos to heifers is generally accepted [23]. Conducting a preventive analysis of target metabolic pathways in the culture media of Grade 2 embryos could optimize overall ET efforts and should be explored in future studies.

Figures and Tables

Figure 1 is updated with “+” or “-“depending on the culture media's metabolite concentration increases or decreases. We decided to update tables with “↓” and “↑” symbols as well, to highlight an increase or a decrease in metabolite concentration in culture media containing embryos compared to the control, which should make reading the tables more convenient.

Reviewer 3 Report

Comments and Suggestions for Authors

The search for new effective criteria for assessing the quality of embryos is currently a very important problem not only in agricultural sciences, but also in reproductive medicine. Although the pre-implantation development of cattle embryos differs significantly from the development of human embryos, the new data presented in the MS under review are, from my point of view, of significant interest to a wide range of specialists. The main goal of the MS was to search for biochemical predictors of the success of implantation of cattle embryos. The authors analyzed the metabolome of culture media in which pre-implantation cattle embryos developed with different implantation outcomes. A number of compounds have been identified (methionine sulfoxide, DOPA, spermidine, etc.), the content of which in the medium varied significantly in the group of successfully implanted embryos. The text is clearly written and the data is discussed sufficiently that I have no major questions or comments after reading the MS.

Minor point

Oocytes were obtained from both pregnant and non-pregnant cows, without the use of hormonal stimulation. Consequently, pregnant and non-pregnant animals had different hormonal status. Could this affect the initial quality of oocytes and the metabolism of embryos in the future? Was the implantation efficiency of embryos obtained from oocytes of pregnant and non-pregnant cows the same? Authors should also indicate in the text how many cows (pregnant and non-pregnant) were used as oocyte donors, how many oocytes were collected, and how many zygotes were obtained after in vitro fertilization.

Author Response

Dear Reviewer,
I appreciate your valuable suggestion and the prompt review of the manuscript. Thank you.

Oocytes were indeed obtained from both pregnant and non-pregnant donor cows, a factor known to influence oocyte quality and blastocyst rates, as indicated in our previous studies (https://onlinelibrary.wiley.com/doi/full/10.1111/rda.13025, http://doi.org/10.1111/rda.13271). Specifically, blastocyst rates are higher when using oocytes from pregnant donor cows. However, in the current study, we did not explore the metabolomic differences of embryos obtained from pregnant and non-pregnant donors. This is an intriguing aspect that we plan to investigate in future studies, as I believe there could be metabolic differences.

We have revised the discussion section of the manuscript accordingly. Please see the attachment. 
